# ISAAC Newton: Input-based Approximate Curvature for Newton's Method

**Felix Petersen**[12], **Tobias Sutter**[2], **Christian Borgelt**[3], **Dongsung Huh**[4],
**Hilde Kuehne**[45], **Yuekai Sun**[6], **Oliver Deussen**[2]
[1]Stanford University, [2]University of Konstanz, [3]University of Salzburg,
[4]MIT-IBM Watson AI Lab, [5]University of Frankfurt, [6]University of Michigan
petersen@cs.stanford.edu

## Abstract

We present ISAAC (Input-baSed ApproximAte Curvature), a novel method that
conditions the gradient using selected second-order information and has an asymp-
totically vanishing computational overhead, assuming a batch size smaller than the
number of neurons. We show that it is possible to compute a good conditioner based
on only the input to a respective layer without a substantial computational over-
head. The proposed method allows effective training even in small-batch stochastic
regimes, which makes it competitive to first-order as well as second-order methods.

## 1 Introduction

While second-order optimization methods are traditionally much less explored than first-order
methods in large-scale machine learning (ML) applications due to their memory requirements and
prohibitive computational cost per iteration, they have recently become more popular in ML mainly
due to their fast convergence properties when compared to first-order methods [1]. The expensive
computation of an inverse Hessian (also known as pre-conditioning matrix) in the Newton step has
also been tackled via estimating the curvature from the change in gradients. Loosely speaking, these
algorithms are known as *quasi-Newton methods*; for a comprehensive treatment, see Nocedal &
Wright [2]. Various approximations to the pre-conditioning matrix have been proposed in recent
literature [3]–[6]. From a theoretical perspective, second-order optimization methods are not nearly
as well understood as first-order methods. It is an active research direction to fill this gap [7], [8].

Motivated by the task of training neural networks, and the observation that invoking local curvature
information associated with neural network objective functions can achieve much faster progress
per iteration than standard first-order methods [9]–[11], several methods have been proposed. One
of these methods, that received significant attention, is known as *Kronecker-factored Approximate
Curvature (K-FAC)* [12], whose main ingredient is a sophisticated approximation to the generalized
Gauss-Newton matrix and the Fisher information matrix quantifying the curvature of the underlying
neural network objective function, which then can be inverted efficiently.

Inspired by the K-FAC approximation and the Tikhonov regularization of the Newton method,
we introduce a novel two parameter regularized Kronecker-factorized Newton update step. The
proposed scheme disentangles the classical Tikhonov regularization and in a specific limit allows
us to condition the gradient using selected second-order information and has an asymptotically
vanishing computational overhead. While this case makes the presented method highly attractive
from the computational complexity perspective, we demonstrate that its empirical performance on
high-dimensional machine learning problems remains comparable to existing SOTA methods.

The contributions of this paper can be summarized as follows: (i) we propose a novel two parameter
regularized K-FAC approximated Gauss-Newton update step; (ii) we prove that for an arbitrary pair of
regularization parameters, the proposed update direction is always a direction of decreasing loss; (iii)
in the limit, as one regularization parameter grows, we obtain an efficient and effective conditioning
of the gradient with an asymptotically vanishing overhead; (iv) we empirically analyze the method
and find that our efficient conditioning method maintains the performance of its more expensive
counterpart; (v) we demonstrate the effectiveness of the method in small-batch stochastic regimes
and observe performance competitive to first-order as well as quasi-Newton methods.

## 2 PRELIMINARIES

In this section, we review aspects of second-order optimization, with a focus on generalized Gauss-Newton methods. In combination with Kronecker factorization, this leads us to a new regularized update scheme. We consider the training of an $L$-layer neural network $f(x; \theta)$ defined recursively as

$$z_i \leftarrow a_{i-1} W^{(i)} \quad \text{(pre-activations)}, \qquad a_i \leftarrow \phi(z_i) \quad \text{(activations)}, \tag{1}$$

where $a_0 = x$ is the vector of inputs and $a_L = f(x; \theta)$ is the vector of outputs. Unless noted otherwise, we assume these vectors to be row vectors (i.e., in $\mathbb{R}^{1 \times n}$) as this allows for a direct extension to the (batch) vectorized case (i.e., in $\mathbb{R}^{b \times n}$) introduced later. For any layer $i$, let $W^{(i)} \in \mathbb{R}^{d_{i-1} \times d_i}$ be a weight matrix and let $\phi$ be an element-wise nonlinear function. We consider a convex loss function $\mathcal{L}(y, y')$ that measures the discrepancy between $y$ and $y'$. The training optimization problem is then

$$\arg \min_\theta \mathbb{E}_{x,y} \left[ \mathcal{L}(f(x; \theta), y) \right], \tag{2}$$

where $\theta = \left[ \theta^{(1)}, \ldots, \theta^{(L)} \right]$ with $\theta^{(i)} = \text{vec}(W^{(i)})$.

The classical Newton method for solving (2) is expressed as the update rule

$$\theta' = \theta - \eta \, \mathbf{H}_\theta^{-1} \nabla_\theta \mathcal{L}(f(x; \theta), y), \tag{3}$$

where $\eta > 0$ denotes the learning rate and $\mathbf{H}_\theta$ is the Hessian corresponding to the objective function in (2). The stability and efficiency of an estimation problem solved via the Newton method can be improved by adding a Tikhonov regularization term [13] leading to a regularized Newton method

$$\theta' = \theta - \eta \left( \mathbf{H}_\theta + \lambda \mathbf{I} \right)^{-1} \nabla_\theta \mathcal{L}(f(x; \theta), y), \tag{4}$$

where $\lambda > 0$ is the so-called Tikhonov regularization parameter. It is well-known [14], [15], that under the assumption of approximating the model $f$ with its first-order Taylor expansion, the Hessian corresponds with the so-called generalized Gauss-Newton (GGN) matrix $\mathbf{G}_\theta$, and hence (4) can be expressed as

$$\theta' = \theta - \eta \left( \mathbf{G}_\theta + \lambda \mathbf{I} \right)^{-1} \nabla_\theta \mathcal{L}(f(x; \theta), y). \tag{5}$$

A major practical limitation of (5) is the computation of the inverse term. A method that alleviates this difficulty is known as Kronecker-Factored Approximate Curvature (K-FAC) [12] which approximates the block-diagonal (i.e., layer-wise) empirical Hessian or GGN matrix. Inspired by K-FAC, there have been other works discussing approximations of $\mathbf{G}_\theta$ and its inverse [15]. In the following, we discuss a popular approach that allows for (moderately) efficient computation.

The generalized Gauss-Newton matrix $\mathbf{G}_\theta$ is defined as

$$\mathbf{G}_\theta = \mathbb{E} \left[ (\mathbf{J}_\theta f(x; \theta))^\top \nabla_f^2 \mathcal{L}(f(x; \theta), y) \, \mathbf{J}_\theta f(x; \theta) \right], \tag{6}$$

where $\mathbf{J}$ and $\nabla^2$ denote the Jacobian and Hessian matrices, respectively. Correspondingly, the diagonal block of $\mathbf{G}_\theta$ corresponding to the weights of the $i$th layer $W^{(i)}$ is

$$\mathbf{G}_{W^{(i)}} = \mathbb{E} \left[ (\mathbf{J}_{W^{(i)}} f(x; \theta))^\top \nabla_f^2 \mathcal{L}(f(x; \theta), y) \, \mathbf{J}_{W^{(i)}} f(x; \theta) \right].$$

According to the backpropagation rule $\mathbf{J}_{W^{(i)}} f(x; \theta) = \mathbf{J}_{z_i} f(x; \theta) \, a_{i-1}$, $a^\top b = a \otimes b$, and the mixed-product property, we can rewrite $\mathbf{G}_{W^{(i)}}$ as

$$\mathbf{G}_{W^{(i)}} = \mathbb{E} \left[ \left( (\mathbf{J}_{z_i} f(x; \theta) \, a_{i-1})^\top (\nabla_f^2 \mathcal{L}(f(x; \theta), y))^{1/2} \right) \left( (\nabla_f^2 \mathcal{L}(f(x; \theta), y))^{1/2} \, \mathbf{J}_{z_i} f(x; \theta) \, a_{i-1} \right) \right] \tag{7}$$

$$= \mathbb{E} \left[ (\bar{g}^\top a_{i-1})^\top (\bar{g}^\top a_{i-1}) \right] = \mathbb{E} \left[ (\bar{g} \otimes a_{i-1})^\top (\bar{g} \otimes a_{i-1}) \right] = \mathbb{E} \left[ (\bar{g}^\top \bar{g}) \otimes (a_{i-1}^\top a_{i-1}) \right], \tag{8}$$

where

$$\bar{g} = (\mathbf{J}_{z_i} f(x; \theta))^\top \, (\nabla_f^2 \mathcal{L}(f(x; \theta), y))^{1/2}. \tag{9}$$

**Remark 1** (Monte-Carlo Low-Rank Approximation for $\bar{g}^\top \bar{g}$)**.** *As $\bar{g}$ is a matrix of shape $m \times d_i$ where $m$ is the dimension of the output of $f$, $\bar{g}$ is generally expensive to compute. Therefore, [12] use a low-rank Monte-Carlo approximation to estimate $\nabla_f^2 \mathcal{L}(f(x; \theta), y)$ and thereby $\bar{g}^\top \bar{g}$. For this, we need to use the distribution underlying the probabilistic model of our loss $\mathcal{L}$ (e.g., Gaussian for MSE loss, or a categorical distribution for cross entropy). Specifically, by sampling from this distribution*

$p_f(x)$ defined by the network output $f(x;\theta)$, we can get an estimator of $\nabla_f^2 \mathcal{L}(f(x;\theta), y)$ via the identity

$$\nabla_f^2 \mathcal{L}(f(x;\theta), y) = \mathbb{E}_{\hat{y} \sim p_f(x)} \left[ \nabla_f \mathcal{L}(f(x;\theta), \hat{y})^\top \nabla_f \mathcal{L}(f(x;\theta), \hat{y}) \right]. \tag{10}$$

An extensive reference for this (as well as alternatives) can be found in Appendix A.2 of Dangel et al. [15]. The respective rank-1 approximation (denoted by $\triangleq$) of $\nabla_f^2 \mathcal{L}(f(x;\theta))$ is

$$\nabla_f^2 \mathcal{L}(f(x;\theta), y) \triangleq \nabla_f \mathcal{L}(f(x;\theta), \hat{y})^\top \nabla_f \mathcal{L}(f(x;\theta), \hat{y}),$$

where $\hat{y} \sim p_f(x)$. Respectively, we can estimate $\bar{g}^\top \bar{g}$ using this rank-1 approximation with

$$\bar{g} \triangleq (\mathbf{J}_{z_i} f(x;\theta))^\top \nabla_f \mathcal{L}(f(x;\theta), \hat{y}) = \nabla_{z_i} \mathcal{L}(f(x;\theta), \hat{y}). \tag{11}$$

In analogy to $\bar{g}$, we introduce the gradient of training objective with respect to pre-activations $z_i$ as

$$\mathrm{g}_i = (\mathbf{J}_{z_i} f(x;\theta))^\top \nabla_f \mathcal{L}(f(x;\theta), y) = \nabla_{z_i} \mathcal{L}(f(x;\theta), y). \tag{12}$$

In other words, for a given layer, let $\mathrm{g} \in \mathbb{R}^{1 \times d_i}$ denote the gradient of the loss between an output and the ground truth and let $\bar{g} \in \mathbb{R}^{m \times d_i}$ denote the derivative of the network $f$ times the square root of the Hessian of the loss function (which may be approximated according to Remark 1), each of them with respect to the output $z_i$ of the given layer $i$. Note that $\bar{g}$ is not equal to $\mathrm{g}$ and that they require one backpropagation pass each (or potentially many for the case of $\bar{g}$). This makes computing $\bar{g}$ costly.

Applying the K-FAC [12] approximation to (8) the expectation of Kronecker products can be approximated as the Kronecker product of expectations as

$$\mathbf{G} = \mathbb{E}((\bar{g}^\top \bar{g}) \otimes (\mathrm{a}^\top \mathrm{a})) \approx \mathbb{E}(\bar{g}^\top \bar{g}) \otimes \mathbb{E}(\mathrm{a}^\top \mathrm{a}), \tag{13}$$

where, for clarity, we drop the index of $\mathrm{a}_{i-1}$ in (8) and denote it with $\mathrm{a}$; similarly we denote $\mathbf{G}_{W^{(i)}}$ as $\mathbf{G}$. While the expectation of Kronecker products is generally not equal to the Kronecker product of expectations, this K-FAC approximation (13) has been shown to be fairly accurate in practice and to preserve the "coarse structure" of the GGN matrix [12]. The K-FAC decomposition in (13) is convenient as the Kronecker product has the favorable property that for two matrices $A, B$ the identity $(A \otimes B)^{-1} = A^{-1} \otimes B^{-1}$ which significantly simplifies the computation of an inverse.

In practice, $\mathbb{E}(\bar{g}^\top \bar{g})$ and $\mathbb{E}(\mathrm{a}^\top \mathrm{a})$ can be computed by averaging over a batch of size $b$ as

$$\mathbb{E}(\bar{g}^\top \bar{g}) \simeq \bar{\mathbf{g}}^\top \bar{\mathbf{g}}/b, \qquad \mathbb{E}(\mathrm{a}^\top \mathrm{a}) \simeq \mathbf{a}^\top \mathbf{a}/b, \tag{14}$$

where we denote batches of $\mathrm{g}$, $\bar{g}$ and $\mathrm{a}$, as $\mathbf{g} \in \mathbb{R}^{b \times d_i}$, $\bar{\mathbf{g}} \in \mathbb{R}^{rb \times d_i}$ and $\mathbf{a} \in \mathbb{R}^{b \times d_{i-1}}$, where our layer has $d_{i-1}$ inputs, $d_i$ outputs, $b$ is the batch size, and $r$ is either the number of outputs $m$ or the rank of an approximation according to Remark 1. Correspondingly, the K-FAC approximation of the GGN matrix and its inverse are concisely expressed as

$$\mathbf{G} \approx (\bar{\mathbf{g}}^\top \bar{\mathbf{g}}) \otimes (\mathbf{a}^\top \mathbf{a})/b^2 \qquad \mathbf{G}^{-1} \approx (\bar{\mathbf{g}}^\top \bar{\mathbf{g}})^{-1} \otimes (\mathbf{a}^\top \mathbf{a})^{-1} \cdot b^2. \tag{15}$$

Equipped with the standard terminology and setting, we now introduce the novel, regularized update step. First, inspired by the K-FAC approximation (13), the Tikhonov regularized Gauss-Newton method (5) can be approximated by

$$\theta^{(i)\prime} = \theta^{(i)} - \eta (\bar{\mathbf{g}}^\top \bar{\mathbf{g}}/b + \lambda \mathbf{I})^{-1} \otimes (\mathbf{a}^\top \mathbf{a}/b + \lambda \mathbf{I})^{-1} \nabla_{\theta^{(i)}} \mathcal{L}(f(x;\theta)), \tag{16}$$

with regularization parameter $\lambda > 0$. A key observation, which is motivated by the structure of the above update, is to disentangle the two occurrences of $\lambda$ into two independent regularization parameters $\lambda_{\mathbf{g}}, \lambda_{\mathbf{a}} > 0$. By defining the Kronecker-factorized Gauss-Newton update step as

$$\boldsymbol{\zeta} = \lambda_{\mathbf{g}} \lambda_{\mathbf{a}} (\bar{\mathbf{g}}^\top \bar{\mathbf{g}}/b + \lambda_{\mathbf{g}} \mathbf{I})^{-1} \otimes (\mathbf{a}^\top \mathbf{a}/b + \lambda_{\mathbf{a}} \mathbf{I})^{-1} \nabla_{\theta^{(i)}} \mathcal{L}(f(x;\theta)), \tag{17}$$

we obtain the concise update equation

$$\theta^{(i)\prime} = \theta^{(i)} - \eta^* \boldsymbol{\zeta}. \tag{18}$$

This update (18) is equivalent to update (16) when in the case of $\eta^* = \frac{\eta}{\lambda_{\mathbf{g}} \lambda_{\mathbf{a}}}$ and $\lambda = \lambda_{\mathbf{g}} = \lambda_{\mathbf{a}}$. This equivalence does not restrict $\eta^*, \lambda_{\mathbf{g}}, \lambda_{\mathbf{a}}$ in any way, and changing $\lambda_{\mathbf{g}}$ or $\lambda_{\mathbf{a}}$ does not mean that we change our learning rate or step size $\eta^*$. Parameterizing $\boldsymbol{\zeta}$ in (17) with the multiplicative terms $\lambda_{\mathbf{g}} \lambda_{\mathbf{a}}$ makes the formulation more convenient for analysis.

In this paper, we investigate the theoretical and empirical properties of the iterative update rule (18) and in particular show how the regularization parameters $\lambda_{\mathbf{g}}, \lambda_{\mathbf{a}}$ affect the Kronecker-factorized Gauss-Newton update step $\boldsymbol{\zeta}$. When analyzing the Kronecker-factorized Gauss-Newton update step $\boldsymbol{\zeta}$, a particularly useful tool is the vector product identity,

$$\left(\left(\bar{\boldsymbol{g}}^\top \bar{\boldsymbol{g}}\right)^{-1} \otimes \left(\mathbf{a}^\top \mathbf{a}\right)^{-1}\right) \mathrm{vec}(\mathbf{g}^\top \mathbf{a}) = \mathrm{vec}\left(\left(\bar{\boldsymbol{g}}^\top \bar{\boldsymbol{g}}\right)^{-1} \mathbf{g}^\top \mathbf{a} \left(\mathbf{a}^\top \mathbf{a}\right)^{-1}\right), \qquad (19)$$

where the gradient with respect to the weight matrix is $\mathbf{g}^\top \mathbf{a}$.

## 3 Theoretical Guarantees

In this section, we investigate the theoretical properties of the Kronecker-factorized Gauss-Newton update direction $\boldsymbol{\zeta}$ as defined in (17). We recall that $\boldsymbol{\zeta}$ introduces a Tikonov regularization, as it is commonly done in implementations of second order-based methods. Not surprisingly, we show that by decreasing the regularization parameters $\lambda_{\mathbf{g}}, \lambda_{\mathbf{a}}$ the update rule (18) collapses (in the limit) to the classical Gauss-Newton method, and hence in the regime of small $\lambda_{\mathbf{g}}, \lambda_{\mathbf{a}}$ the variable $\boldsymbol{\zeta}$ describes the Gauss-Newton direction. Moreover, by increasing the regularization strength, we converge (in the limit) to the conventional gradient descent update step.

The key observation is that, as we disentangle the regularization of the two Kronecker factors $\bar{\boldsymbol{g}}^\top \bar{\boldsymbol{g}}$ and $\mathbf{a}^\top \mathbf{a}$, and consider the setting where only one regularizer is large ($\lambda_{\mathbf{g}} \to \infty$ to be precise), we obtain an update direction that can be computed highly efficiently. We show that this setting describes an approximated Gauss-Newton update scheme, whose superior numerical performance is then empirically demonstrated in Section 4.

**Theorem 1** (Properties of $\boldsymbol{\zeta}$). *The K-FAC based update step $\boldsymbol{\zeta}$ as defined in* (17) *can be expressed as*

$$\boldsymbol{\zeta} = \left(\mathbf{I}_m - \frac{1}{b\lambda_{\mathbf{g}}}\bar{\boldsymbol{g}}^\top \left(\mathbf{I}_b + \frac{1}{b\lambda_{\mathbf{g}}}\bar{\boldsymbol{g}}\bar{\boldsymbol{g}}^\top\right)^{-1}\bar{\boldsymbol{g}}\right) \cdot \mathbf{g}^\top \cdot \left(\mathbf{I}_b - \frac{1}{b\lambda_{\mathbf{a}}}\mathbf{a}\mathbf{a}^\top \left(\mathbf{I}_b + \frac{1}{b\lambda_{\mathbf{a}}}\mathbf{a}\mathbf{a}^\top\right)^{-1}\right) \cdot \mathbf{a}. \tag{20}$$

*Moreover, $\boldsymbol{\zeta}$ admits the following asymptotic properties:*

*(i) In the limit of $\lambda_{\mathbf{g}}, \lambda_{\mathbf{a}} \to 0$, $\frac{1}{\lambda_{\mathbf{g}}\lambda_{\mathbf{a}}}\boldsymbol{\zeta}$ is the K-FAC approximation of the Gauss-Newton step, i.e., $\lim_{\lambda_{\mathbf{g}}, \lambda_{\mathbf{a}} \to 0} \frac{1}{\lambda_{\mathbf{g}}\lambda_{\mathbf{a}}}\boldsymbol{\zeta} \approx \mathbf{G}^{-1}\nabla_{\theta^{(i)}}\mathcal{L}(f(x; \theta))$, where $\approx$ denotes the K-FAC approximation* (15).*

*(ii) In the limit of $\lambda_{\mathbf{g}}, \lambda_{\mathbf{a}} \to \infty$, $\boldsymbol{\zeta}$ is the gradient, i.e., $\lim_{\lambda_{\mathbf{g}}, \lambda_{\mathbf{a}} \to \infty} \boldsymbol{\zeta} = \nabla_{\theta^{(i)}}\mathcal{L}(f(x; \theta))$.*

*The Proof is deferred to the Supplementary Material.*

We want to show that $\boldsymbol{\zeta}$ is well-defined and points in the correct direction, not only for $\lambda_{\mathbf{g}}$ and $\lambda_{\mathbf{a}}$ numerically close to zero because we want to explore the full spectrum of settings for $\lambda_{\mathbf{g}}$ and $\lambda_{\mathbf{a}}$. Thus, we prove that $\boldsymbol{\zeta}$ is a direction of increasing loss, independent of the choices of $\lambda_{\mathbf{g}}$ and $\lambda_{\mathbf{a}}$.

**Theorem 2** (Correctness of $\boldsymbol{\zeta}$ is independent of $\lambda_{\mathbf{g}}$ and $\lambda_{\mathbf{a}}$). *$\boldsymbol{\zeta}$ is a direction of increasing loss, independent of the choices of $\lambda_{\mathbf{g}}$ and $\lambda_{\mathbf{a}}$.*

*Proof.* Recall that $(\lambda_{\mathbf{g}}\mathbf{I}_m + \bar{\boldsymbol{g}}^\top \bar{\boldsymbol{g}}/b)$ and $(\lambda_{\mathbf{a}}\mathbf{I}_n + \mathbf{a}^\top \mathbf{a}/b)$ are positive semi-definite (PSD) matrices by definition. Their inverses $(\lambda_{\mathbf{g}}\mathbf{I}_m + \bar{\boldsymbol{g}}^\top \bar{\boldsymbol{g}}/b)^{-1}$ and $(\lambda_{\mathbf{a}}\mathbf{I}_n + \mathbf{a}^\top \mathbf{a}/b)^{-1}$ are therefore also PSD. As the Kronecker product of PSD matrices is PSD, the conditioning matrix $((\lambda_{\mathbf{g}}\mathbf{I}_m + \bar{\boldsymbol{g}}^\top \bar{\boldsymbol{g}}/b)^{-1} \otimes (\lambda_{\mathbf{a}}\mathbf{I}_n + \mathbf{a}^\top \mathbf{a}/b)^{-1} \approx \mathbf{G}^{-1})$ is PSD, and therefore the direction of the update step remains correct. $\square$

*This leads us to our primary contribution:* From our formulation of $\boldsymbol{\zeta}$, we can find that, in the limit for $\lambda_{\mathbf{g}} \to \infty$, Equation (21) does not depend on $\bar{\boldsymbol{g}}$. This is computationally very beneficial as computing $\bar{\boldsymbol{g}}$ is costly as it requires one or even many additional backpropagation passes. In addition, it allows conditioning the gradient update by multiplying a $b \times b$ matrix between $\mathbf{g}^\top$ and $\mathbf{a}$, which is very fast.

**Theorem 3** (Efficient Update Direction / ISAAC). *In the limit of $\lambda_{\mathbf{g}} \to \infty$, the update step $\boldsymbol{\zeta}$ converges to $\lim_{\lambda_{\mathbf{g}} \to \infty} \boldsymbol{\zeta} = \boldsymbol{\zeta}^*$, where*

$$\boldsymbol{\zeta}^* = \mathbf{g}^\top \cdot \left(\mathbf{I}_b - \frac{1}{b\lambda_{\mathbf{a}}}\mathbf{a}\mathbf{a}^\top \left(\mathbf{I}_b + \frac{1}{b\lambda_{\mathbf{a}}}\mathbf{a}\mathbf{a}^\top\right)^{-1}\right) \cdot \mathbf{a}. \tag{21}$$

*(i) Here, the update direction $\boldsymbol{\zeta}^*$ is based only on the inputs and does not require computing $\bar{\boldsymbol{g}}$ (which would require a second backpropagation pass), making it efficient.*

*(ii) The computational cost of computing the update $\boldsymbol{\zeta}^*$ lies in $\mathcal{O}(bn^2 + b^2n + b^3)$, where $n$ is the number of neurons in each layer. This comprises the conventional cost of computing the gradient $\nabla = \mathbf{g}^\top \mathbf{x}$ lying in $\mathcal{O}(bn^2)$, and the overhead of computing $\boldsymbol{\zeta}^*$ instead of $\nabla$ lying in $\mathcal{O}(b^2n + b^3)$. The overhead is vanishing, assuming $n \gg b$. For $b > n$ the complexity lies in $\mathcal{O}(bn^2 + n^3)$.*

*Proof.* We first show the property (21). Note that according to (22), $\lambda_{\mathbf{g}} \cdot \left(\lambda_{\mathbf{g}}\mathbf{I}_m + \bar{\boldsymbol{g}}^\top\bar{\boldsymbol{g}}/b\right)^{-1}$ converges in the limit of $\lambda_{\mathbf{g}} \to \infty$ to $\mathbf{I}_m$, and therefore (21) holds.

(i) The statement follows from the fact that the term $\bar{\boldsymbol{g}}$ does not appear in the equivalent characterization (21) of $\boldsymbol{\zeta}^*$.

(ii) We first note that the matrix $\mathbf{a}\mathbf{a}^\top$ is of dimension $b \times b$, and can be computed in $\mathcal{O}(b^2n)$ time. Next, the matrix

$$\left(\mathbf{I}_b - \frac{1}{b\lambda_{\mathbf{a}}}\mathbf{a}\mathbf{a}^\top \left(\mathbf{I}_b + \frac{1}{b\lambda_{\mathbf{a}}}\mathbf{a}\mathbf{a}^\top\right)^{-1}\right)$$

is of shape $b \times b$ and can be multiplied with $\mathbf{a}$ in $\mathcal{O}(b^2n)$ time. $\qquad\square$

Notably, (21) can be computed with a vanishing computational overhead and with only minor modifications to the implementation. Specifically, only the $\mathbf{g}^\top\mathbf{a}$ expression has to be replaced by (21) in the backpropagation step. As this can be done independently for each layer, this lends itself also to applying it only to individual layers.

As we see in the experimental section, in many cases in the mini-batch regime (i.e., $b < n$), the optimal (or a good) choice for $\lambda_{\mathbf{g}}$ actually lies in the limit to $\infty$. This is a surprising result, leading to the efficient and effective $\boldsymbol{\zeta}^* = \boldsymbol{\zeta}_{\lambda_{\mathbf{g}} \to \infty}$ optimizer.

**Remark 2** (Relation between Update Direction $\boldsymbol{\zeta}$ and $\boldsymbol{\zeta}^*$)**.** *When comparing the update direction $\boldsymbol{\zeta}$ in (20) without regularization (i.e., $\lambda_{\mathbf{g}} \to 0, \lambda_{\mathbf{a}} \to 0$) with $\boldsymbol{\zeta}^*$ (i.e., $\lambda_{\mathbf{g}} \to \infty$) as given in (21), it can be directly seen that $\boldsymbol{\zeta}^*$ corresponds to a particular pre-conditioning of $\boldsymbol{\zeta}$, since $\boldsymbol{\zeta}^* = M\boldsymbol{\zeta}$ for $M = \frac{1}{b\lambda_{\mathbf{g}}}\bar{\boldsymbol{g}}^\top\bar{\boldsymbol{g}}$.*

As the last theoretical property of our proposed update direction $\boldsymbol{\zeta}^*$, we show that in specific networks $\boldsymbol{\zeta}^*$ coincides with the Gauss-Newton update direction.

**Theorem 4** ($\boldsymbol{\zeta}^*$ is Exact for the Last Layer)**.** *For the case of linear regression or, more generally, the last layer of networks, with the mean squared error, $\boldsymbol{\zeta}^*$ is the Gauss-Newton update direction.*

*Proof.* The Hessian matrix of the mean squared error loss is the identity matrix. Correspondingly, the expectation value of $\bar{\boldsymbol{g}}^\top\bar{\boldsymbol{g}}$ is $\mathbf{I}$. Thus, $\boldsymbol{\zeta}^* = \boldsymbol{\zeta}$. $\qquad\square$

**Remark 3.** *The direction $\boldsymbol{\zeta}^*$ corresponds to the Gauss-Newton update direction with an approximation of $\mathbf{G}$ that can be expressed as $\mathbf{G} \approx \mathbb{E}\left[\mathbf{I} \otimes (\mathrm{a}^\top\mathrm{a})\right]$.*

**Remark 4** (Extension to the Natural Gradient)**.** *In some cases, it might be more desirable to use the Fisher-based natural gradient instead of the Gauss-Newton method. The difference to this setting is that in (5) the GGN matrix $\mathbf{G}$ is replaced by the empirical Fisher information matrix $\mathbf{F}$.*

*We note that our theory also applies to $\mathbf{F}$, and that $\boldsymbol{\zeta}^*$ also efficiently approximates the natural gradient update step $\mathbf{F}^{-1}\nabla$. The $i$-th diagonal block of $\mathbf{F}$ ($\mathbf{F}_{\theta^{(i)}} = \mathbb{E}\left[(\mathrm{g}_i^\top\mathrm{g}_i) \otimes (a_{i-1}^\top a_{i-1})\right]$), has the same form as a block of the GGN matrix $\mathbf{G}$ ($\mathbf{G}_{\theta^{(i)}} = \mathbb{E}\left[(\bar{g}_i^\top\bar{g}_i) \otimes (a_{i-1}^\top a_{i-1})\right]$). Thus, we can replace $\bar{\boldsymbol{g}}$ with $\mathbf{g}$ in our theoretical results to obtain their counterparts for $\mathbf{F}$.*

## 4 EXPERIMENTS[1]

In the previous section, we discussed the theoretical properties of the proposed update directions $\boldsymbol{\zeta}$ and $\boldsymbol{\zeta}^*$ with the aspect that $\boldsymbol{\zeta}^*$ would actually be "free" to compute in the mini-batch regime. In this section, we provide empirical evidence that $\boldsymbol{\zeta}^*$ is a good update direction, even in deep learning. Specifically, we demonstrate that

---

[1]Code will be made available at github.com/Felix-Petersen/isaac

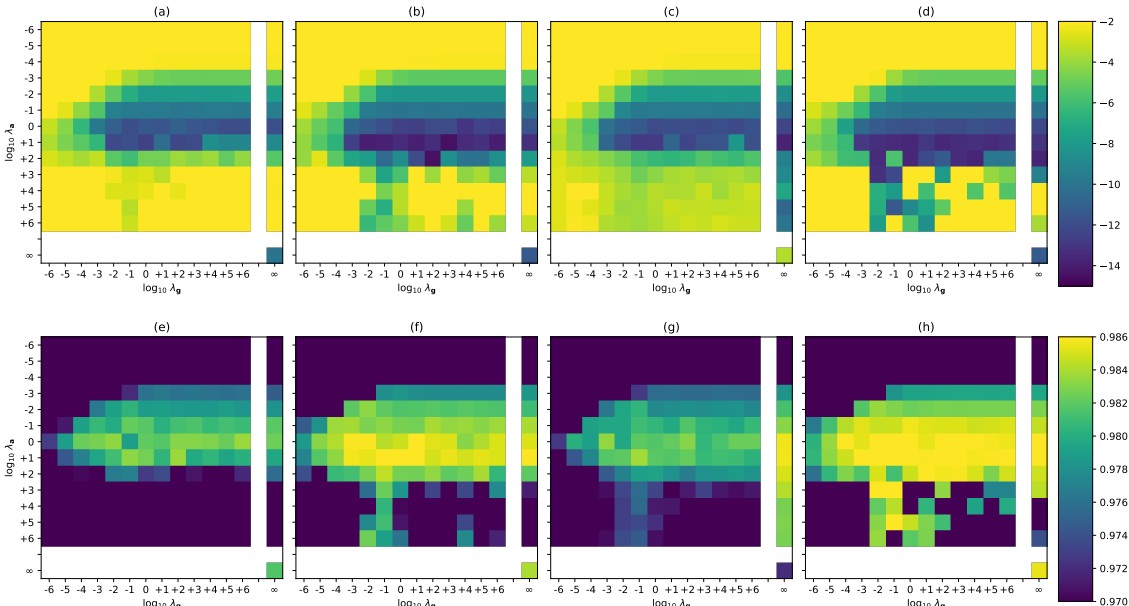

Figure 1: Logarithmic training loss (top) and test accuracy (bottom) on the MNIST classification task. The axes are the regularization parameters $\lambda_{\mathbf{g}}$ and $\lambda_{\mathbf{a}}$ in logarithmic scale with base 10. Training with a 5-layer ReLU activated network with 100 (left, a, e), 400 (center, b, c, f, g), and 1 600 (right, d, h) neurons per layer. The optimizer is SGD except for (c, g) where the optimizer is SGD with momentum. The top-left sector is $\boldsymbol{\zeta}$, the top-right column is $\boldsymbol{\zeta}^*$, and the bottom-right corner is $\nabla$ (gradient descent). For each experiment and each of the three sectors, we use one learning rate, i.e., $\boldsymbol{\zeta}, \boldsymbol{\zeta}^*, \nabla$ have their own learning rate to make a fair comparison between the methods; within each sector the learning rate is constant. We can observe that in the limit of $\lambda_{\mathbf{g}} \to \infty$ (i.e., in the limit to the right) the performance remains good, showing the utility of $\boldsymbol{\zeta}^*$.

(E1) $\boldsymbol{\zeta}^*$ achieves similar performance to K-FAC, while being substantially cheaper to compute.

(E2) The performance of our proposed method can be empirically maintained in the mini-batch regime ($n \gg b$).

(E3) $\boldsymbol{\zeta}^*$ may be used for individual layers, while for other layers only the gradient $\nabla$ is used. This still leads to improved performance.

(E4) $\boldsymbol{\zeta}^*$ also improves the performance for training larger models such as BERT and ResNet.

(E5) The runtime and memory requirements of $\boldsymbol{\zeta}^*$ are comparable to those of gradient descent.

E1: IMPACT OF REGULARIZATION PARAMETERS

For (E1), we study the dependence of the model's performance on the regularization parameters $\lambda_{\mathbf{g}}$ and $\lambda_{\mathbf{a}}$. Here, we train a 5-layer deep neural network on the MNIST classification task [16] with a batch size of 60 for a total of 40 epochs or 40 000 steps.

The plots in Figure 1 demonstrate that the advantage of training by conditioning with curvature information can be achieved by considering both layer inputs $\mathbf{a}$ and gradients with respect to random samples $\bar{\mathbf{g}}$, but also using only layer inputs $\mathbf{a}$. In the plot, we show the performance of $\boldsymbol{\zeta}$ for different choices of $\lambda_{\mathbf{g}}$ and $\lambda_{\mathbf{a}}$, each in the range from $10^{-6}$ to $10^6$. The right column shows $\boldsymbol{\zeta}^*$, i.e., $\lambda_{\mathbf{g}} = \infty$, for different $\lambda_{\mathbf{a}}$. The bottom-right corner is gradient descent, which corresponds to $\lambda_{\mathbf{g}} = \infty$ and $\lambda_{\mathbf{a}} = \infty$.

Newton's method or the general K-FAC approximation corresponds to the area with small $\lambda_{\mathbf{g}}$ and $\lambda_{\mathbf{a}}$. The interesting finding here is that the performance does not suffer by increasing $\lambda_{\mathbf{g}}$ toward $\infty$, i.e., from left to right in the plot.

In addition, in Figure 3, we consider the case of regression with an auto-encoder trained with the MSE loss on MNIST [16] and Fashion-MNIST [17]. Here, we follow the same principle as above and also find that $\boldsymbol{\zeta}^*$ performs well.

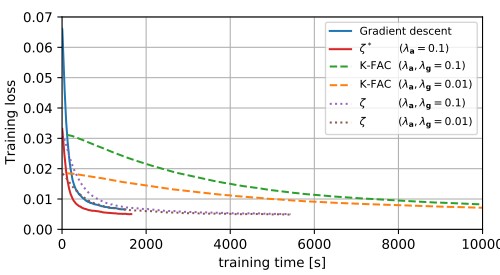
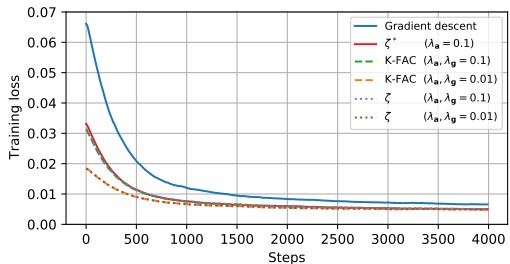

Figure 2: Training loss of the MNIST auto-encoder trained with gradient descent, K-FAC, $\zeta$, and $\zeta^*$. Comparing the performance per real-time (left) and per number of update steps (right). Runtimes are for a CPU core.

In Figure 2, we compare the loss for different methods. Here, we distinguish between loss per time (left) and loss per number of steps (right). We can observe that, for $\lambda = 0.1$, K-FAC, $\zeta$, and $\zeta^*$ are almost identical per update step (right), while $\zeta^*$ is by a large margin the fastest, followed by $\zeta$, and the conventional K-FAC implementation is the slowest (left). On the other hand, for $\lambda = 0.01$ we can achieve a faster convergence than with $\lambda = 0.1$, but here only the K-FAC and $\zeta$ methods are numerically stable, while $\zeta^*$ is unstable in this case. This means in the regime of very small $\lambda$, $\zeta^*$ is not as robust as K-FAC and $\zeta$, however, it achieves good performance with small but moderate $\lambda$ like $\lambda = 0.1$. For $\lambda < 0.01$, also K-FAC and $\zeta$ become numerically unstable in this setting and, in general, we observed that the smallest valid $\lambda$ for K-FAC is 0.01 or 0.001 depending on model and task. Under consideration of the runtime, $\zeta^*$ performs best as it is almost as fast as gradient descent while performing equivalent to K-FAC and $\zeta$. Specifically, a gradient descent step is only about 10% faster than $\zeta^*$.

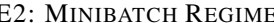

Figure 3: Training an auto-encoder on MNIST (left) and Fashion-MNIST (right). The model is the same as used by Botev *et al.* [18], i.e., it is a ReLU-activated 6-layer fully connected model with dimensions `784-1000-500- 30-500-1000-784`. Displayed is the logarithmic training loss.

Figure 4: Training a 5-layer ReLU network with 400 neurons per layer on the MNIST classification task (as in Figure 1) but with the Adam optimizer [19].

### E2: MINIBATCH REGIME

For (E2), in Figure 1, we can see that training performs well for $n \in \{100, 400, 1\,600\}$ neurons per layer at a batch size of only 60. Also, in all other experiments, we use small batch sizes of between 8 and 100.

### E3: $\zeta^*$ IN INDIVIDUAL LAYERS

In Figure 5, we train the 5-layer fully connected model with 400 neurons per layer. Here, we consider the setting that we use $\zeta^*$ in some of the layers while using the default gradient $\nabla$ in other layers. Specifically, we consider the

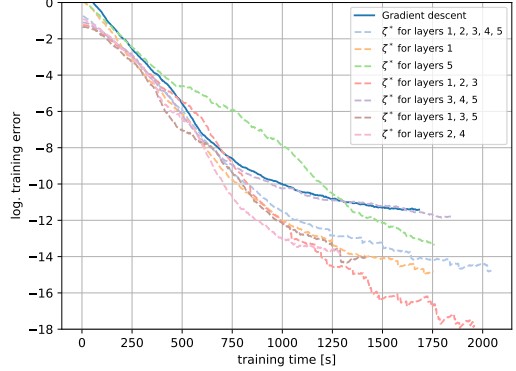

Figure 5: Training on the MNIST classification task using $\zeta^*$ only in selected layers. Runtimes are for CPU.

settings, where all, the first, the final, the first three, the final three, the odd numbered, and the even numbered layers are updated by $\zeta^*$. We observe that all settings with $\zeta^*$ perform better than

Table 1: BERT results for fine-tuning pre-trained BERT-Base (B-B) and BERT-Mini (B-M) models on the COLA, MRPC, and STSB text classification tasks. Larger values are better for all metrics. MCC is the Matthews correlation. Results averaged over 10 runs.

| Method / Setting | CoLA (B-B) | CoLA (B-M) | MRPC (B-B) | | STS-B (B-M) | |
| --- | --- | --- | --- | --- | --- | --- |
| Metric | MCC | MCC | Acc. | F1 | Pearson | Spearman |
| Gradient baseline | $54.20 \pm 7.56$ | $21.08 \pm 2.88$ | $82.52 \pm 1.22$ | $87.88 \pm 0.74$ | $76.98 \pm 1.10$ | $76.88 \pm 0.79$ |
| $\zeta^*$ | $57.62 \pm 1.59$ | $24.67 \pm 2.62$ | $83.28 \pm 0.89$ | $88.28 \pm 0.70$ | $81.09 \pm 1.58$ | $80.82 \pm 1.57$ |

plain gradient descent, except for "$\zeta^*$ for layers 3,4,5" which performs approximately equivalent to gradient descent.

### E4: LARGE-SCALE MODELS

**BERT** To demonstrate the utility of $\zeta^*$ also in large-scale models, we evaluate it for fine-tuning BERT [20] on three natural language tasks. In Table 1, we summarize the results for the BERT fine-tuning task. For the "Corpus of Linguistic Acceptability" (CoLA) [21] data set, we fine-tune both the BERT-Base and the BERT-Mini models and find that we outperform the gradient descent baseline in both cases. For the "Microsoft Research Paraphrase Corpus" (MRPC) [22] data set, we fine-tune the BERT-Base model and find that we outperform the baseline both in terms of accuracy and F1-score. Finally, on the "Semantic Textual Similarity Benchmark" (STS-B) [23] data set, we fine-tune the BERT-Mini model and achieve higher Pearson and Spearman correlations than the baseline. While for training with CoLA and MRPC, we were able to use the Adam optimizer [19] (which is recommended for this task and model) in conjunction with $\zeta^*$ in place of the gradient, for STS-B Adam did not work well. Therefore, for STS-B, we evaluated it using the SGD with momentum optimizer. For each method, we performed a grid search over the hyperparameters. We note that we use a batch size of $8$ in all BERT experiments.

**ResNet** In addition, we conduct an experiment where we train the last layer of a ResNet with $\zeta^*$, while the remainder of the model is updated using the gradient $\nabla$. Here, we train a ResNet-18 [24] on CIFAR-10 [25] using SGD with a batch size of $100$. In Figure 6, we plot the test accuracy against number of epochs. The times for each method lie within $1\%$ of each other. We consider three settings: the typical setting with momentum and weight decay, a setting with only momentum, and a setting with vanilla SGD without momentum. The results show that the proposed method outperforms SGD in each of these cases. While the improvements are rather small in the case of the default training, they are especially large in the case of no weight decay and no momentum.

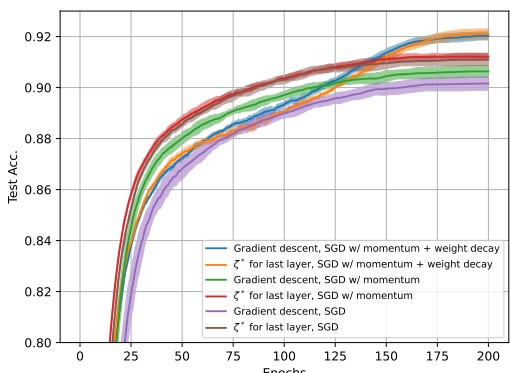

Figure 6: ResNet-18 trained on CIFAR-10 with image augmentation and a cosine learning rate schedule. To ablate the optimizer, two additional settings are added, specifically, without weight decay and without momentum. Results are averaged over 5 runs and the standard deviation is indicated with the colored areas.

### E5: RUNTIME AND MEMORY

Finally, we also evaluate the runtime and memory requirements of each method. The runtime evaluation is displayed in Table 2. We report both CPU and GPU runtime using PyTorch [26] and (for K-FAC) the backpack library [15]. Note that the CPU runtime is more representative of the pure computational cost, as for the first rows of the GPU runtime the overhead of calling the GPU is dominant. When comparing runtimes between the gradient and $\zeta^*$ on the GPU, we can observe that we have an overhead of around $2.5\,s$ independent of the model size. The overhead for CPU time is also very small at less than $1\%$ for the largest model, and only $1.3\,s$ for the smallest model. In contrast, the runtime of $\zeta^*$ is around $4$ times the runtime of the gradient, and K-FAC has an even substantially larger runtime. Regarding memory, $\zeta^*$ (contrasting the other approaches) also requires only a small additional footprint.

Table 2: Runtimes and memory requirements for different models. Runtime is the training time per epoch on MNIST at a batch size of 60, i.e., for 1 000 training steps. The K-FAC implementation is from the `backpack` library [15]. The GPU is an Nvidia A6000.

| Model | Gradient | | | K-FAC | | | $\zeta$ | | | $\zeta^*$ | | |
|---|---|---|---|---|---|---|---|---|---|---|---|---|
| | CPU time | GPU time | Memory | CPU time | GPU t. | Memory | CPU time | GPU t. | Memory | CPU t. | GPU t. | Memory |
| 5 layers w/ 100 n. | 2.05 s | 1.79 s | 1.0 MB | 62.78 s | 17.63 s | 11.5 MB | 8.65 s | 11.76 s | 1.6 MB | 3.34 s | 4.07 s | 1.0 MB |
| 5 layers w/ 400 n. | 23.74 s | 1.84 s | 4.8 MB | 218.48 s | 32.00 s | 22.4 MB | 38.67 s | 12.62 s | 7.7 MB | 13.62 s | 4.19 s | 4.9 MB |
| 5 layers w/ 1 600 n. | 187.87 s | 1.93 s | 51.0 MB | 6985.48 s | 156.48 s | 212.2 MB | 665.80 s | 12.53 s | 85.8 MB | 291.01 s | 4.49 s | 51.4 MB |
| 5 layers w/ 6 400 n. | 3439.59 s | 8.22 s | 691.0 MB | — | 1320.81 s | 3155.3 MB | 9673 s | 31.87 s | 1197.8 MB | 3451.61 s | 10.24 s | 692.5 MB |
| Auto-Encoder | 78.61 s | 2.20 s | 16.2 MB | 1207.58 s | 74.09 s | 70.7 MB | 193.25 s | 14.19 s | 33.8 MB | 87.39 s | 4.93 s | 16.5 MB |

**Remark 5** (Implementation). *The implementation of $\zeta^*$ can be done by replacing the backpropagation step of a respective layer by (21). As all "ingredients" are already available in popular deep learning frameworks, it requires only little modification (contrasting K-FAC and $\zeta$, which require at least one additional backpropagation.)*

We will publish the source code of our implementation. In the appendix, we give a PyTorch [26] implementation of the proposed method ($\zeta^*$).

## 5 RELATED WORK

Our methods are related to K-FAC by Martens and Grosse [12]. K-FAC uses the approximation (13) to approximate the blocks of the Hessian of the empirical risk of neural networks. In most implementations of K-FAC, the off-diagonal blocks of the Hessian are also set to zero. One of the main claimed benefits of K-FAC is its speed (compared to stochastic gradient descent) for large-batch size training. That said, recent empirical work has shown that this advantage of K-FAC disappears once the additional computational costs of hyperparameter tuning for large batch training is accounted for. There is a line of work that extends the basic idea of K-FAC to convolutional layers [27]. Botev *et al.* [18] further extend these ideas to present KFLR, a Kronecker factored low-rank approximation, and KFRA, a Kronecker factored recursive approximation of the Gauss-Newton step. Singh and Alistarh [28] propose WoodFisher, a Woodbury matrix inverse-based estimate of the inverse Hessian, and apply it to neural network compression. Yao *et al.* [29] propose AdaHessian, a second-order optimizer that incorporates the curvature of the loss function via an adaptive estimation of the Hessian. Frantar *et al.* [6] propose M-FAC, a matrix-free approximation of the natural gradient through a queue of the (e.g., 1 000) recent gradients. These works fundamentally differ from our approach in that their objective is to approximate the Fisher or Gauss-Newton matrix inverse vector products. In contrast, this work proposes to approximate the Gauss-Newton matrix by only one of its Kronecker factors, which we find to achieve good performance at a substantial computational speedup and reduction of memory footprint. For an overview of this area, we refer to Kunstner *et al.* [30] and Martens [31]. For an overview of the technical aspects of backpropagation of second-order quantities, we refer to Dangel *et al.* [15], [32]

Taking a step back, K-FAC is one of many Newton-type methods for training neural networks. Other prominent examples of such methods include subsampled Newton methods [33], [34] (which approximate the Hessian by subsampling the terms in the empirical risk function and evaluating the Hessian of the subsampled terms) and sketched Newton methods [3]–[5] (which approximate the Hessian by sketching, e.g., by projecting the Hessian to a lower-dimensional space by multiplying it with a random matrix). Another quasi-Newton method [35] proposes approximating the Hessian by a block-diagonal matrix using the structure of gradient and Hessian to further approximate these blocks. The main features that distinguish K-FAC from this group of methods are K-FAC's superior empirical performance and K-FAC's lack of theoretical justification.

## 6 CONCLUSION

In this work, we presented ISAAC Newton, a novel approximate curvature method based on layer-inputs. We demonstrated it to be a special case of the regularization-generalized Gauss-Newton method and empirically demonstrate its utility. Specifically, our method features an asymptotically vanishing computational overhead in the mini-batch regime, while achieving competitive empirical performance on various benchmark problems.

## ACKNOWLEDGMENTS

This work was supported by the IBM-MIT Watson AI Lab, the DFG in the Cluster of Excellence EXC 2117 "Centre for the Advanced Study of Collective Behaviour" (Project-ID 390829875), the Land Salzburg within the WISS 2025 project IDA-Lab (20102-F1901166-KZP and 20204-WISS/225/197-2019), and the National Science Foundation (NSF) (grants no. 1916271, 2027737, and 2113373).

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

## A PyTorch Implementation

We display a PyTorch [26] implementation of ISAAC for a fully-connected layer below. Here, we mark the important part (i.e., the part beyond the boilerplate) with a red rectangle.

```python
import torch

class ISAACLinearFunction(torch.autograd.Function):
    @staticmethod
    def forward(ctx, input, weight, bias, la, inv_type):
        ctx.save_for_backward(input, weight, bias)
        ctx.la = la
        if inv_type == 'cholesky_inverse':
            ctx.inverse = torch.cholesky_inverse
        elif inv_type == 'inverse':
            ctx.inverse = torch.inverse
        else:
            raise NotImplementedError(inv_type)
        return input @ weight.T + (bias if bias is not None else 0)

    @staticmethod
    def backward(ctx, grad_output):
        input, weight, bias = ctx.saved_tensors
        if ctx.needs_input_grad[0]:
            grad_0 = grad_output @ weight
        else:
            grad_0 = None

        if ctx.needs_input_grad[1]:

            aaT = input @ input.T / grad_output.shape[0]
            I_b = torch.eye(aaT.shape[0], device=aaT.device, dtype=aaT.dtype)
            aaT_IaaT_inv = aaT @ ctx.inverse(aaT / ctx.la + I_b)
            grad_1 = grad_output.T @ (
                    I_b - 1. / ctx.la * aaT_IaaT_inv
            ) @ input

        else:
            grad_1 = None

        return (
            grad_0,
            grad_1,
            grad_output.mean(0, keepdim=True) if bias is not None else None,
            None, None, None,
        )

class ISAACLinear(torch.nn.Linear):
    def __init__(self, in_features, out_features,
                 la, inv_type='inverse', **kwargs):
        super(ISAACLinear, self).__init__(
            in_features=in_features, out_features=out_features, **kwargs
        )
        self.la = la
        self.inv_type = inv_type

    def forward(self, input: torch.Tensor) -> torch.Tensor:
        return ISAACLinearFunction.apply(
            input, self.weight,
```

```
        self.bias.unsqueeze(0) if self.bias is not None else None,
        self.la,
        self.inv_type
    )
```

## B  IMPLEMENTATION DETAILS

Unless noted differently, for all experiments, we tune the learning rate on a grid of $(1, 0.3, 0.1, 0.03, 0.01, 0.003, 0.001)$. We verified this range to cover the full reasonable range of learning rates. Specifically, for every single experiment, we made sure that there is no learning rate outside this range which performs better.

For all language model experiments, we used the respective Huggingface PyTorch implementation.

All other hyperparameter details are given in the main paper.

## C  ADDITIONAL PROOFS

*Proof of Theorem 1.* We first show, that $\boldsymbol{\zeta}$ as defined in (17) can be expressed as in (20). Indeed by using (19), the Woodbury matrix identity and by regularizing the inverses, we can see that

$$
\begin{aligned}
\boldsymbol{\zeta} &= \lambda_{\mathbf{g}}\lambda_{\mathbf{a}}(\bar{\boldsymbol{g}}^\top\bar{\boldsymbol{g}}/b + \lambda_{\mathbf{g}}\mathbf{I})^{-1} \otimes (\mathbf{a}^\top\mathbf{a}/b + \lambda_{\mathbf{a}}\mathbf{I})^{-1}\mathbf{g}^\top\mathbf{a} \\
&= \lambda_{\mathbf{g}}\lambda_{\mathbf{a}} \cdot \left(\lambda_{\mathbf{g}}\mathbf{I}_m + \bar{\boldsymbol{g}}^\top\bar{\boldsymbol{g}}/b\right)^{-1}\mathbf{g}^\top\mathbf{a}\left(\lambda_{\mathbf{a}}\mathbf{I}_n + \mathbf{a}^\top\mathbf{a}/b\right)^{-1} \\
&= \lambda_{\mathbf{g}}\lambda_{\mathbf{a}} \cdot \left(\frac{1}{\lambda_{\mathbf{g}}}\mathbf{I}_m - \frac{1}{b\lambda_{\mathbf{g}}^2}\bar{\boldsymbol{g}}^\top\left(\mathbf{I}_b + \frac{1}{b\lambda_{\mathbf{g}}}\bar{\boldsymbol{g}}\bar{\boldsymbol{g}}^\top\right)^{-1}\bar{\boldsymbol{g}}\right) \\
&\quad \mathbf{g}^\top\mathbf{a}\left(\frac{1}{\lambda_{\mathbf{a}}}\mathbf{I}_n - \frac{1}{b\lambda_{\mathbf{a}}^2}\mathbf{a}^\top\left(\mathbf{I}_b + \frac{1}{b\lambda_{\mathbf{a}}}\mathbf{a}\mathbf{a}^\top\right)^{-1}\mathbf{a}\right) \\
&= \left(\mathbf{I}_m - \frac{1}{b\lambda_{\mathbf{g}}}\bar{\boldsymbol{g}}^\top\left(\mathbf{I}_b + \frac{1}{b\lambda_{\mathbf{g}}}\bar{\boldsymbol{g}}\bar{\boldsymbol{g}}^\top\right)^{-1}\bar{\boldsymbol{g}}\right) \cdot \mathbf{g}^\top \\
&\quad \cdot \mathbf{a} \cdot \left(\mathbf{I}_n - \frac{1}{b\lambda_{\mathbf{a}}}\mathbf{a}^\top\left(\mathbf{I}_b + \frac{1}{b\lambda_{\mathbf{a}}}\mathbf{a}\mathbf{a}^\top\right)^{-1}\mathbf{a}\right) \\
&= \left(\mathbf{I}_m - \frac{1}{b\lambda_{\mathbf{g}}}\bar{\boldsymbol{g}}^\top\left(\mathbf{I}_b + \frac{1}{b\lambda_{\mathbf{g}}}\bar{\boldsymbol{g}}\bar{\boldsymbol{g}}^\top\right)^{-1}\bar{\boldsymbol{g}}\right) \cdot \mathbf{g}^\top \\
&\quad \cdot \left(\mathbf{a} - \frac{1}{b\lambda_{\mathbf{a}}}\mathbf{a}\mathbf{a}^\top\left(\mathbf{I}_b + \frac{1}{b\lambda_{\mathbf{a}}}\mathbf{a}\mathbf{a}^\top\right)^{-1}\mathbf{a}\right) \\
&= \left(\mathbf{I}_m - \frac{1}{b\lambda_{\mathbf{g}}}\bar{\boldsymbol{g}}^\top\left(\mathbf{I}_b + \frac{1}{b\lambda_{\mathbf{g}}}\bar{\boldsymbol{g}}\bar{\boldsymbol{g}}^\top\right)^{-1}\bar{\boldsymbol{g}}\right) \cdot \mathbf{g}^\top \\
&\quad \cdot \left(\mathbf{I}_b - \frac{1}{b\lambda_{\mathbf{a}}}\mathbf{a}\mathbf{a}^\top\left(\mathbf{I}_b + \frac{1}{b\lambda_{\mathbf{a}}}\mathbf{a}\mathbf{a}^\top\right)^{-1}\right) \cdot \mathbf{a}
\end{aligned}
$$

To show Assertion (i), we note that according to (17)

$$
\begin{aligned}
&\lim_{\lambda_{\mathbf{g}},\lambda_{\mathbf{a}}\to 0} \frac{1}{\lambda_{\mathbf{g}}\lambda_{\mathbf{a}}}\boldsymbol{\zeta} \\
&= \lim_{\lambda_{\mathbf{g}},\lambda_{\mathbf{a}}\to 0} (\bar{\boldsymbol{g}}^\top\bar{\boldsymbol{g}}/b + \lambda_{\mathbf{g}}\mathbf{I})^{-1} \otimes (\mathbf{a}^\top\mathbf{a}/b + \lambda_{\mathbf{a}}\mathbf{I})^{-1}\mathbf{g}^\top\mathbf{a} \\
&= (\bar{\boldsymbol{g}}^\top\bar{\boldsymbol{g}})^{-1} \otimes (\mathbf{a}^\top\mathbf{a})^{-1}\mathbf{g}^\top\mathbf{a} \\
&\approx \mathbf{G}^{-1}\mathbf{g}^\top\mathbf{a},
\end{aligned}
$$

where the first equality uses the definition of $\boldsymbol{\zeta}$ in (17). The second equality is due to the continuity of the matrix inversion and the last approximate equality follows from the K-FAC approximation (15).

To show Assertion (ii), we consider $\lim_{\lambda_{\mathbf{g}}\to\infty}$ and $\lim_{\lambda_{\mathbf{a}}\to\infty}$ independently, that is

$$\lim_{\lambda_{\mathbf{g}}\to\infty} \lambda_{\mathbf{g}} \cdot \left(\lambda_{\mathbf{g}}\mathbf{I}_m + \bar{\mathbf{g}}^\top\bar{\mathbf{g}}/b\right)^{-1} \tag{22}$$
$$= \lim_{\lambda_{\mathbf{g}}\to\infty} \left(\mathbf{I}_m + \frac{1}{b\lambda_{\mathbf{g}}}\bar{\mathbf{g}}^\top\bar{\mathbf{g}}\right)^{-1} = \mathbf{I}_m,$$

and

$$\lim_{\lambda_{\mathbf{a}}\to\infty} \lambda_{\mathbf{a}} \cdot \left(\lambda_{\mathbf{a}}\mathbf{I}_n + \mathbf{a}^\top\mathbf{a}/b\right)^{-1} \tag{23}$$
$$= \lim_{\lambda_{\mathbf{a}}\to\infty} \left(\mathbf{I}_n + \frac{1}{b\lambda_{\mathbf{a}}}\mathbf{a}^\top\mathbf{a}\right)^{-1} = \mathbf{I}_n.$$

This then implies

$$\lim_{\lambda_{\mathbf{g}},\lambda_{\mathbf{a}}\to\infty} \lambda_{\mathbf{g}} \left(\lambda_{\mathbf{g}}\mathbf{I}_m + \bar{\mathbf{g}}^\top\bar{\mathbf{g}}/b\right)^{-1} \cdot \mathbf{g}^\top \tag{24}$$
$$\cdot \mathbf{a} \cdot \lambda_{\mathbf{a}} \left(\lambda_{\mathbf{a}}\mathbf{I}_n + \mathbf{a}^\top\mathbf{a}/b\right)^{-1}$$
$$= \mathbf{I}_m \cdot \mathbf{g}^\top\mathbf{a} \cdot \mathbf{I}_n = \mathbf{g}^\top\mathbf{a},$$

which concludes the proof. $\qquad\square$

## D ADDITIONAL EXPERIMENTS

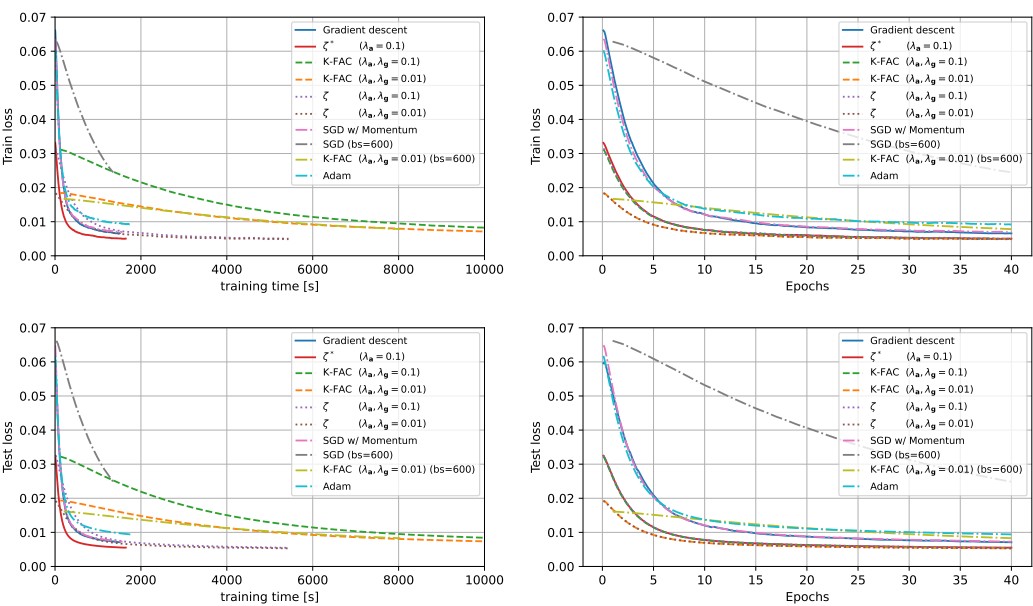

Figure 7: Training loss of the MNIST auto-encoder trained with gradient descent, K-FAC, $\boldsymbol{\zeta}, \boldsymbol{\zeta}^*$, as well as SGD w/ momentum, SGD with a $10\times$ larger batch size (600), K-FAC with a $10\times$ larger batch size (600), and Adam. Comparing the performance per real-time (left) and per number of epochs (right). We display both the training loss (top) as well as the test loss (bottom) Runtimes are for a CPU core.

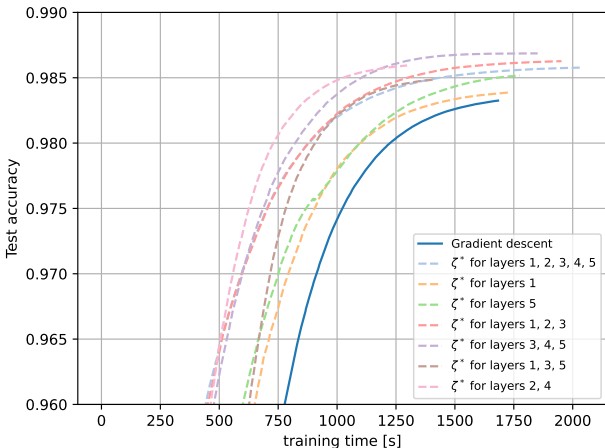

Figure 8: Test accuracy for training on the MNIST classification task using $\boldsymbol{\zeta}^*$ only in selected layers. Runtimes are for CPU.

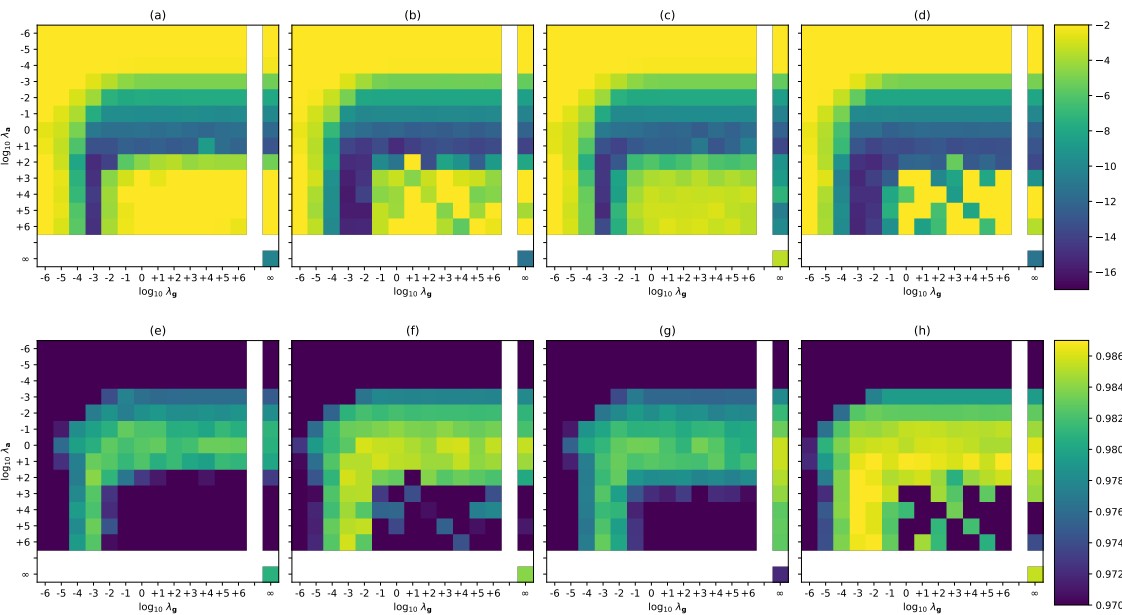

Figure 9: Reproduction of the experiments in Figure 1 but with the Fisher-based natural gradient formulation from Remark 4. For a description of the experimental settings, see the caption of Figure 1. We observe that, for large $\lambda_{\mathbf{g}}$, the behavior is similar to Figure 1, which is expected as they are the same in the limit of $\lambda_{\mathbf{g}} \to \infty$. Further, we observe that (in this case of the Fisher-based $\boldsymbol{\zeta}$) not only in the limit of $\lambda_{\mathbf{g}} \to \infty$ but also in the limit of $\lambda_{\mathbf{a}} \to \infty$ good performance can be achieved. Moreover, in this specific experiment, $\lambda_{\mathbf{a}} \to \infty$ has slightly better optimal performance compared to $\lambda_{\mathbf{g}} \to \infty$, but $\lambda_{\mathbf{a}} \to \infty$ is more sensitive to changes in $\lambda_g$ compared to the sensitivity of the case of $\lambda_{\mathbf{g}} \to \infty$ wrt. changes in $\lambda_{\mathbf{a}}$. This phenomenon was also (to a lesser extent) visible in the experiments of Figure 1. We would like to remark that the case of $\lambda_{\mathbf{g}} \to \infty$ (i.e., $\boldsymbol{\zeta}^\star$) is computationally more efficient compared to $\lambda_{\mathbf{a}} \to \infty$.

