# OpenReview forum: "ISAAC Newton: Input-based Approximate Curvature for Newton's Method"
_ICLR.cc/2023/Conference — ICLR 2023 poster_

### Official Review · Reviewer_2oTp · 2022-10-22

**Confidence:** 3
**Correctness:** 4
**Technical Novelty And Significance:** 2
**Empirical Novelty And Significance:** 2
**Recommendation:** 5

**Clarity, Quality, Novelty And Reproducibility:**

Overall the paper is well written.

However, the contribution is mostly adding ridge penalties, which seems rather incremental.

**Strength And Weaknesses:**

* Strength

The paper shows that their proposed method works well in practice.

* Weakness
- There are rather few experiments
- The main change in the paper is adding ridge penalties, which seems rather incremental

**Summary Of The Paper:**

The paper presents a method that conditions the gradient using selected second-order information, which uses less computation than naively running Newton.

**Summary Of The Review:**

The paper seems like a minor modification of K-FAC by adding some stochasticity and ridge penalties.

---

### Official Review · Reviewer_xNi3 · 2022-10-24

**Confidence:** 2
**Clarity, Quality, Novelty And Reproducibility:** This paper is easy to follow.
**Correctness:** 3
**Technical Novelty And Significance:** 2
**Empirical Novelty And Significance:** 2
**Recommendation:** 6

**Strength And Weaknesses:**

This paper proposes novel second-order algorithms for training neural networks.
The experiments show the proposed algorithms are computation efficient.


**Summary Of The Paper:**

This paper seems only adding a $\lambda  \cdot I$ to the KFAC.
The theory in this paper seems only an easy extension of the work of KFAC.
The most useful part of this paper is the case $\lambda_g \to \infty$.
This paper gives a descent direction without the expensive cost of computing $\bar{g}$.
The experiments show the proposed algorithms are computation efficient.

**Summary Of The Review:**

This paper seems only adding a $\lambda  \cdot I$ to the KFAC.
The most useful part of this paper is the case $\lambda_g \to \infty$ which avoids the expensive cost of computing $\bar{g}$.
The experiments show the proposed algorithms are computation efficient.

I am not very familiar with training algorithms of  neural networks.
Thus, I am not sure the value of the proposed algorithms.

---

### Official Review · Reviewer_U2Re · 2022-11-04

**Confidence:** 5
**Correctness:** 3
**Technical Novelty And Significance:** 4
**Empirical Novelty And Significance:** 3
**Recommendation:** 8

**Clarity, Quality, Novelty And Reproducibility:**

### Quality
Most of the technical sections of the paper are well written. Surprisingly, I found that the part that requires re-visiting is the Preliminaries section, especially the first page before Remark 1.
Specifically:
 * I had a hard time in the beginning training my mental model to treat $\mathbf{\alpha}^\top \mathbf{\alpha}$ as a Kronecker outer product, instead of the standard inner product. I get that this is the result of treating all vectors as row vectors, but it still requires mental effort to adjust to the new reality.
 * I do not understand the reasons behind the constant change in notation for the Hessian, we have seen it denoted as $\mathbf{H}$ and also as $\nabla^2$, which is a symbol normally used for the Laplacian.
 * The in-line math text just before Eq. (7) is problematic. The Jacobian is a matrix with shape $m \times d_i$, while the shape of $\alpha_{i-1}$ is a row vector of $1 \times d_{i-1}$. This two objects cannot be multiplied together to get the overall Jacobian, $J_{z_i}$. You need a Kronecker product there to get the correct shape $m \times d_{i}d_{i-1}$. Obviously Eq.(7) and (8) need revisiting.
 * In Eq. (8), the final term should have $\alpha_{i-1}^\top \alpha_{i-1}$ and not $\alpha_{i-1}^\top \otimes \alpha_{i-1}$. The Kronecker product should not be there.
* In Eq. (17), why do we have the term $\lambda_g \lambda_a$ in the front?

Apart from the above, I believe the remaining is of high quality. Theorems 1&3 are nice additions to the literature.
Regarding the experiments, I am satisfied with the evaluation and I believe the results suggest that there is value in the proposed approximate Gauss-Newton step, since it can reach better performances in various tasks with a little computational overhead (big plus for the wall clock experiment).

* Minor issue here, in the beginning of page 7, the text that supports Figure 2, wrongly references Figure 7 (which is in the supplementary material).

One extra point that I want to raise is that although I found Remark 4 (the connection to natural gradient) very interesting, I certainly missed an experiment comparing the two. It would definitely made a nice addition and would complete the analysis.


### Clarity
The paper is very well written. Normally manuscripts on second order optimisation methods tend to be intimidating. The authors here have done a really good job on navigating the reader all the way through the derivations.


**Strength And Weaknesses:**

The paper explores a new pathway to approximate a Gauss-Newton step. I found this idea and the analysis simple and novel enough. The related work is sufficiently covered and, the presented approach is nicely motivated from the well studied K-FAC and the differences with which the authors intend to attack the problem.

See next section for some of my questions/issues I raised.


**Summary Of The Paper:**

The paper presents a new approximation to the Gauss-Newton update step, in similar fashion to the well studied Kronecker-factored approximate curvature (K-FAC). The authors start from the generalised Gauss-Newton matrix $\mathbf{G}_\theta$ and follow a Monte-Carlo low-rank approximation to it before applying the K-FAC to approximate the expectation of the Kronecker with the Kronecker of the expectations.
Then, they introduce independent Tikhonov regularisation to each of the terms in the Kronecker. By taking the limit of one of the regularisations to infinity, they prove that the "expensive" gradient term vanishes and the resulting gradient step is just a preconditioned gradient that accounts for the curvature of the space. The authors have experimentally evaluated the proposed update step in various set-ups, proving the effectiveness and the efficacy of it.


**Summary Of The Review:**

In my opinion, the paper provides a good theoretical result and also a nice practical optimisation step which can be used as an easy approximation to second order optimisation.

---

### Decision · Program_Chairs · 2023-01-20

**Decision:**

Accept: poster

**Justification For Why Not Higher Score:**

The experiments are really very weak for a deep-learning paper. Experiments solely on MNIST are nearly useless in the context of larger models, especially when it comes to second-order methods.

**Justification For Why Not Lower Score:**

See above. In dubio pro reo.

**Metareview: Summary, Strengths And Weaknesses:**

This paper proposes an algorithmic extension to KFAC. The novelty of the contribution is debatable, and the experiment results are very limited (MNIST only, which is essentially not acceptable anymore in deep learning research). However, the reviewers have found some good things to say about this paper. In particular, Reviewer U2Re sees some novel insights in the construction of the regulariser (see below). The other two reviewers were completely unreachable during the discussion phase, and I believe this should not count against the authors of this paper. I thus consider this paper borderline, but acceptable.


**Note From Pc:**

if the above contains the word "oral" or "spotlight" please see: "oral" presentation means -> notable-top-5% and "spotlight" means -> notable-top-25%. As stated in our emails, we are disassociating presentation type from AC recommendations

**Summary Of Ac-Reviewer Meeting:**

I reached out repeatedly to all reviewers. Only reviewer U2Re responded. It was thus not possible to set up a meeting, but Reviewer U2Re
 provided further input, arguing strongly in favour of this paper. I quote:

> I still believe strongly that this is a good paper and a nice addition to the literature. In my experience there is no such a thing as “trivial extension”. Everything is an extension to something and everything is trivial once someone shows it to you. Having said that, my opinion is that the assigned paper is elegant, simple and to the point. These are some quality attributes that you can’t easily find in many publications lately. There is a good motivation behind introducing the regulariser and the study around $\zeta^*$ and the connection to $\lambda_g \to\infty$ is a nice observation.